# Carpal Tunnel Syndrome Automated Diagnosis: A Motor vs. Sensory Nerve Conduction-Based Approach

**DOI:** 10.3390/bioengineering11020175

**Published:** 2024-02-11

**Authors:** Dimitrios Bakalis, Prokopis Kontogiannis, Evangelos Ntais, Yannis V. Simos, Konstantinos I. Tsamis, George Manis

**Affiliations:** 1Department of Computer Science and Engineering, School of Engineering, University of Ioannina, 45110 Ioannina, Greece; 2Department of Neurology, University Hospital of Ioannina, 45110 Ioannina, Greece; 3Department of Physiology, Faculty of Medicine, University of Ioannina, 45110 Ioannina, Greece

**Keywords:** carpal tunnel syndrome, nerve conduction studies, machine learning, deep learning, multilevel wavelet decomposition, short-time Fourier transform, spectrogram

## Abstract

The objective of this study was to evaluate the effectiveness of machine learning classification techniques applied to nerve conduction studies (NCS) of motor and sensory signals for the automatic diagnosis of carpal tunnel syndrome (CTS). Two methodologies were tested. In the first methodology, motor signals recorded from the patients’ median nerve were transformed into time-frequency spectrograms using the short-time Fourier transform (STFT). These spectrograms were then used as input to a deep two-dimensional convolutional neural network (CONV2D) for classification into two categories: patients and controls. In the second methodology, sensory signals from the patients’ median and ulnar nerves were subjected to multilevel wavelet decomposition (MWD), and statistical and non-statistical features were extracted from the decomposed signals. These features were utilized to train and test classifiers. The classification target was set to three categories: normal subjects (controls), patients with mild CTS, and patients with moderate to severe CTS based on conventional electrodiagnosis results. The results of the classification analysis demonstrated that both methodologies surpassed previous attempts at automatic CTS diagnosis. The classification models utilizing the motor signals transformed into time-frequency spectrograms exhibited excellent performance, with average accuracy of 94%. Similarly, the classifiers based on the sensory signals and the extracted features from multilevel wavelet decomposition showed significant accuracy in distinguishing between controls, patients with mild CTS, and patients with moderate to severe CTS, with accuracy of 97.1%. The findings highlight the efficacy of incorporating machine learning algorithms into the diagnostic processes of NCS, providing a valuable tool for clinicians in the diagnosis and management of neuropathies such as CTS.

## 1. Introduction

The current clinical practice in the management of carpal tunnel syndrome (CTS) focuses on the use of confirmatory diagnostic tests, like ultrasound and nerve conduction studies (NCS), before decision making on further treatment options [1]. In this way, the management of this common medical condition, with an undemanding and quick clinical diagnosis, is significantly prolonged and its cost is rising, especially if we take into account that these tests are not always available in local medical facilities. It is thus pertinent to seek methods of easier diagnosis for this disorder [2]. The main symptoms that characterize the syndrome are sensory deficits and pain in the median nerve allocation area, while weakness implicates the late stages of CTS. These symptoms arise from the compression of the median nerve at the wrist level and the NCS has to reveal the resulting mononeuropathy in order to confirm the diagnosis.

Various studies have employed machine learning (ML) models for the detection and monitoring of carpal tunnel syndrome (CTS). These models leverage diverse datasets, encompassing MRI and CT images, alongside clinical and historical information. For instance, reference [3] applied a deep learning (DL) model to identify ultrasound (US) image features for CTS diagnosis, achieving notable performance with accuracy and recall of 0.94 and 0.95%, respectively. An other investigation [4,5] utilized a convolutional neural network (CNN) model for CTS diagnosis based on CT scan images, attaining accuracy of 0.94%, through validation with images from 53 patients. Furthermore, in a previous study of our group [6], we used machine learning techniques and analyzed the features of NCS sensory signals for the median nerve and managed to reach accuracy of 91.1% for the diagnosis of CTS.

Despite these achievements, certain limitations characterize these studies: (1) a reliance on costly MRI and CT scan images, which may not always be readily available for diagnosis; (2) insufficient consideration of the overlap of CTS symptoms with other conditions, such as cervical radiculopathy, de Quervain tendinopathy, and peripheral neuropathy; and (3) the neglect of essential clinical, personal, and historical data, which significantly impact both disease diagnosis and treatment outcomes.

In this paper, the analysis of two distinct signal types, namely motor and sensory, was deemed essential for two primary reasons. Firstly, it was imperative to develop an automated procedure applicable irrespective of the signal type extracted by a future physician. Secondly, the differentiation between motor and sensory signals necessitated a dual experimental approach, combining artificial intelligence with signal processing methodologies. This comprehensive strategy aimed to enhance the generalizability of the automated diagnosis of CTS and underscores the significance of a multidimensional investigative framework in medical research.

In the first proposed methodology, the NCS classification method was based on a deep two-dimensional convolutional neural network (CONV2D), utilizing motor signals extracted from the participant’s median nerve (compound motor nerve action potential, CMAP). The one-dimensional NCS motor signals were transformed into two-dimensional time-frequency spectrograms, using the short-time Fourier transform (STFT), in order to produce a highly sensitive image dataset for the training and testing of the CONV2D. Motor signals contain a comparatively lower amount of information than sensory signals [7], which renders the classification process quite challenging. However, these signals comprise significantly less noise than sensory signals. Thus, it is important to devise an efficient diagnostic protocol for CTS that can be carried out on motor signals.

In the second proposed methodology, the diagnosis of the mononeuropathy was based on information from the sensory signals of both the median and the ulnar nerves (sensory nerve action potential, SNAP). We applied multilevel wavelet decomposition (MWD), with the level of depth being 5, and we calculated the statistical and non-statistical features from both nerves. Combining all features, a unique dataset was created, suitable for the training and testing of any machine learning classification algorithm. In contrast to other neuropathies that can affect both the median and ulnar nerves, in CTS, only the median nerve is compressed, leaving the ulnar nerve completely unaffected, as shown in Figure 1. The combined information extracted from both nerves is critical, not only in diagnosing CTS but also in ruling out any other possible nerve diseases.

## 2. Materials and Methods

### 2.1. Data Acquisition and Selection

For this study, we used both motor and sensory sets of signal data, from the median and the ulnar nerves, in order to examine the diagnostic ability of our two proposed methodologies. Additionally, signals from both hands of several participants were utilized in this study, raising the potential for bias in the statistical analysis. However, the exceptional performance of the algorithms in the classification procedure suggests that any potential issues arising from this aspect were negligible.

The training dataset used in our initial work [6] was employed to train our models as it contained signals cautiously selected with low noise and controlled diagnostic procedures. It consisted of 28 patients with symptoms and signs, in either one or both hands, suggestive of CTS, and 10 age-matched healthy individuals. The duration of each sensory signal, from both the median and ulnar nerve, was 6 milliseconds, with a total of 1200 discrete voltage measurements. On the other hand, the duration of each motor signal extracted from the median nerve was 6 milliseconds, with a total of 2400 discrete voltage measurements. The initial set was split into two training datasets in order to test both methodologies. In the first training set, the available sensory NCS signals were split into three groups, based on the electrodiagnosis results and according to a widely accepted neurophysiological grading scale [8]. The first group was designated the control group without CTS (*non*), the second group was designated the mild CTS group (*mi*), and the third group was designed the moderate to severe CTS group (*m*/*s*). For the *non-CTS* group, we had 19 hands; for the *mi-CTS* group, we had 21 hands; and for the *m*/*s-CTS* group, we had 36 hands.

In order to increase our training samples, we duplicated all signals and added Gaussian noise [9] to each duplicate, with a noise threshold of 0.8. This methodology not only doubled our initial training dataset but also added some challenging cases that the model could potentially face during the evaluation runs, when using the raw validation data. The resulting set can be seen in Table 1.

In the second training dataset, we split the available motor signals into two distinct categories: *Control* and *Patient*. In the *Control* category, the set had 19 hands, and in the *Patient* category, the set included the remaining 57 hands. Additionally, we created duplicates from all signals and added the same Gaussian noise to double the training dataset, as can be seen in Table 2.

The validation dataset, which was also used in a recently presented abstract [10], was employed to validate the efficacy of the proposed methodologies in the analysis of real-world recordings characterized by a significantly higher level of noise compared to the recordings in the preceding dataset. This dataset originated from a retrospective analysis of 301 randomly selected individuals that were examined with NCS on the upper extremities during the last two years at the Laboratory of Clinical Neurophysiology, Neurological Department of the University Hospital of Ioannina. Each subject diagnosed with the condition exhibited symptoms in either one or both hands.

The integration of sensory signal data derived from the median and ulnar nerves resulted in the compilation of a composite dataset comprising 139 hand samples. Instances lacking data from both nerves were excluded from the sensory nerve methodology. Subsequently, the signals were categorized into three distinct groups: the *non-CTS* group encompassing 51 samples, the *mi-CTS* group comprising 44 samples, and the *m*/*s-CTS* group including 44 samples, as delineated in Table 3.

In contrast, to test our second proposed methodology, we extracted the motor signals that were derived from the median nerve from the aforementioned 301 individuals. As a consequence of the underlying condition whereby a motor signal contains less information than a sensory signal and as the differences between patient groups were insignificant, we opted to divide them into two categories, transforming the diagnosis procedure into a binary classification problem. A representation of two different motor signal waveforms is given in Figure 2. For our analysis, we had available data from 243 right hands and 74 left hands. Subsequently, the signals were categorized into two new distinct categories: the *Control* category, the previous *non-CTS* group, having 172 samples; and the *Patient* category, the previous *mi-CTS* and *m*/*s-CTS* groups combined, including 145 samples, as delineated in Table 4.

### 2.2. Motor Nerve Conduction Approach

#### 2.2.1. Method Overview

The yielded NCS motor signals were divided into data recordings with an identical duration of 6 ms. Each signal was transformed into an image of a time-frequency spectrogram by using the STFT, and the NCS spectrogram images were fed into a custom CONV2D. With the obtained NCS spectrogram images, the classification of the two NCS types was performed in the CONV2D classifier automatically and intelligently.

#### 2.2.2. Motor Signal Pre-Processing

To facilitate the utilization of the proposed CONV2D model, the input data had to be formatted as images. Therefore, in order to incorporate the motor signals into the model, they needed to be transformed into an appropriate image format. Thus, the NCS signals were transformed into 2D time-frequency spectrogrms using the STFT. The STFT is a Fourier-related transformation used to determine the sinusoidal frequency and phase content of local sections of a signal, as it changes over time [11]. In practice, the procedure to compute STFTs is to divide a longer time signal into shorter segments of equal length and then compute the Fourier transform separately on each shorter segment. This produces the Fourier spectrum on each shorter segment. In the discrete time case, the data to be transformed could be broken up into chunks or frames (which usually overlapped each other), to reduce artifacts at the boundary. On each chunk, the Fourier transform was applied and the complex result was added to a matrix, which stored the magnitude and phase for each point, in time and frequency. This procedure can be expressed as
(1)$$STFT\left\{x\left[n\right]\right\}\left(m,\omega \right)\equiv X\left(\tau ,\omega \right)=\sum_{n=-\infty }^{\infty }x\left[n\right]w{\left[n-m\right]}^{e^{-j\omega n}},$$
where x[n] is the nerve signal and w[n] is the window function—in our case, the Hann Window [12]. In this study, we used 20 equal frames with a window size of 120 milivolt values. The squared magnitude of the STFT yielded the spectrogram representation of the power spectral density as ${\left|X\left(\tau ,\omega \right)\right|}^{2}$. Therefore, we transformed the NCS time domain signals into NCS spectrum images by plotting each NCS data recording as an individual 256 × 256 pixel image. A sample of each class’s spectrogram is shown in Figure 3.

#### 2.2.3. Convolutional Neural Network

A CNN was adopted as the NCS nerve classifier. The CNN was first introduced by LeCun [13] and was developed through a project to recognize handwritten zip codes. A CONV2D model can extract the correlation between spatially adjacent pixels, as well as various local features, by using multiple nonlinear filters.

In the initial hidden layer, we implemented a Convolutional2D layer composed of 16 kernels, each with a size of 4 × 4. The activation function employed in this and every following layer was the rectified linear unit (ReLU) [14]. Subsequently, a MaxPooling2D layer with a pool size of 3 × 3 was incorporated. As a result, the output shape of the first layer was determined to be 84 × 84 × 16. In the second hidden layer, we once more implemented a Convolution2D layer composed of 32 kernels, each with a size of 4 × 4. Subsequently, a MaxPooling2D layer with a pool size of 3 × 3 was incorporated, while the output shape of the second layer was 27 × 27 × 32. In the third hidden layer, we implemented a final Convolution2D layer composed of 64 kernels, each with a size of 4 × 4. Next, a MaxPooling2D with a pool size of 2 × 2 was added, while the output shape of the third layer was 8 × 8 × 64. Afterwards, we used a flatten layer, followed by a dense layer consisting of 256 nodes. Between the dense and flatten layers, we used the dropout method to reduce the number of computations in each epoch as a regularization technique. Finally, we added the last layer of the architecture, consisting of one node, utilizing the sigmoid activation function.

The loss metric is defined as the difference between the predicted value of the model and the true value for a specific sample. This metric has several distinct types of mathematical expressions. In this study, we chose the function of binary cross-entropy loss. The loss can be expressed as a formula of the form
(2)$$Loss=-\frac{1}{N}\sum_{i=1}^{N}y_{i}\cdot \log\left(p\left(y_{i}\right)\right)+\left(1-y_{i}\right)\cdot \log\left(1-p\left(y_{i}\right)\right),$$
with *y* being the label and p(y) being the predicted probability of the point being positive for all N points. A visual representation of the CONV2D architecture can be seen in Figure 4. Additionally, we provide some explanations for the layers, as well as the activation functions used in the proposed model, in Table 5.

#### 2.2.4. Model Parameter Optimization

In this proposed COVN2D model, we chose to adjust two main parameters, the batch size and the learning rate. In order to achieve the best classification performance on NCS nerve abnormalities, the step of model parameter optimization was indispensable. To evaluate the importance of the learning rate within the model, we conducted a series of experiments with different parameter sets. We tested the model with different learning rates, while keeping the value of the batch size at 3. We set the number of iteration steps at 30. Figure 5 presents the accuracy and loss value curves for seven different values of the learning rate. Based on the graphical representations, it can be deduced that the optimal learning rate for the model was determined to be 0.005. This particular value exhibited a consistently stable pattern, leading to convergence towards accuracy of 1 and loss of 0.

### 2.3. Sensory Nerve Conduction Approach

#### 2.3.1. Method Overview

For this methodology, we opted to combine information from both the ulnar and median nerves utilizing the sensory signal from each participant’s hand. In particular, for each individual participant, we had two signals with a duration of 6 milliseconds each and 1200 discrete millivolt values. In order to decompose the signals, we used multilevel discrete wavelet decomposition (MWD), with a depth level of 5 scales for the median nerve and a depth level of 2 scales for the ulnar nerve. The main information was stored in the decomposition of the median nerve, with the information from the ulnar nerve being supplementary to our analysis. For the approximate as well as the detail coefficients, we calculated 9 statistical features and 3 non-statistical ones for each sub-signal. This methodology created a set of 108 combined features for each entry in the dataset. Finally, we split the dataset into training and testing sets for several machine learning models, suitable for multi-classification problems.

#### 2.3.2. Multilevel Discrete Wavelet Decomposition

The Hungarian mathematician Alfréd Haar [15] proposed the first discrete wavelet transform. The Haar wavelet transform can be used to pair up input values, storing the difference and passing the sum, for an input represented by a list of $2^{n}$ numbers. It is also considered as a series of rescaled *square-shaped* functions that form a wavelet family or basis. Wavelet analysis, like Fourier analysis, allows for the representation of a target function over an interval in terms of an orthonormal basis. This process is repeated recursively, with the sums paired up to prove the next scale, resulting in $2^{n}-1$ differences and a final sum. The MWD [16] method extracts multilevel time-frequency features from a time series by decomposing the series into low- and high-frequency sub-series, level by level, as shown in Figure 6. The low-pass filter as well as the high-pass filter used in MWD with the Haar wavelet method can be expressed as a formula of the form
(3)$$f_{low}^{z+1}=\frac{x^{z}\left[i-1\right]+x^{z}\left[i\right]}{2},$$
(4)$$f_{high}^{z+1}=\frac{x^{z}\left[i-1\right]-x^{z}\left[i\right]}{2},$$
with *i* taking values from (1,…,n) and *z* being the depth of decomposition.

#### 2.3.3. Feature Extraction

With the approximate as well as the detail coefficients calculated, we can extract various statistical and non-statistical features from the sub-signals. In this work, a comprehensive and systematic approach was taken to identify the most effective features for the task at hand. After evaluating over 50 of the most commonly used features in the literature, the best-performing features were selected through a rigorous experimentation process, which involved applying statistical tests such as chi-squared [17], ANOVA [18] and *t*-tests [19], while evaluating the model’s performance using different evaluation metrics. The results of this process showed that the chosen features were the most suitable for the specific problem, as they provided the best fit and produced the highest performance in comparison to other features. The exclusion of highly correlated features was critical to ensuring the independence and distinctiveness of the chosen features, thereby improving the robustness and accuracy of the models. Additionally, to reduce the computing complexity, we sought to use as few features as possible. We used 12 statistical features and 3 non-statistical features, in order to extract any valuable information, while keeping the total number of features relatively small. Specifically, the statistical features were the mean, standard deviation, min, max, root mean squared, median, skewness, kurtosis, 5th percentile, 25th percentile, 75th percentile, and 95th percentile. The 3 non-statistical features were zero crossing indices, mean crossing indices, and Rényi entropy [20]. The MWD, when applied on the median sensory signal with a depth of 5, returned 5 detail coefficients and 1 approximate coefficient. We extracted the mentioned features from each coefficient, resulting in a total of 90 features from a single signal. Additionally, we applied the MWD, on the ulnar sensory signal with a depth of 2, returning 2 detail coefficients and 1 approximate coefficient. Using the same technique, we extracted 45 features from the coefficients. The total number of extracted features for one patient was 135 in total, with information from both the median and ulnar nerves, as intended.

#### 2.3.4. Machine Learning Algorithms

The diverse range of machine learning models employed in this study provided a robust analysis of the classification problem of CTS. Linear models, such as logistic regression, offer interpretability and simplicity, which can be advantageous for straightforward classification tasks with a limited number of features. On the other hand, nonlinear models, such as support vector machines and neural networks, can capture more complex relationships between the input variables and the output classes and thus are better suited for more challenging classification tasks. In this study, we employed 10 different machine learning models to be able to evaluate the performance of each model and compare their results to select the most suitable models for the diagnosis of CTS. In terms of simplicity, we selected the 5 best-performing models, which were gradient boosting (GB) [21], multi-layer perceptron (MLP) [22], random forest (RF) [23], logistic regression (LR) [24], and K-nearest neighbors (K-NN) [25].

We attempted to use the same parameters for all models, whenever possible. Most machine learning models were trained for 100 epochs, using mini-batches of size 32, with the root mean squared error (RMSE) being the considered loss. For each model, we employed a systematic hyperparameter optimization technique to ascertain the optimal configuration. Specifically, we employed the methodology of grid search cross-validation (GridSearchCV). GridSearchCV entailed an exhaustive search over a predefined hyperparameter grid that allowed us to comprehensively explore the parameter space of each model. By systematically evaluating various hyperparameter combinations, GridSearchCV facilitated the identification of the most suitable parameter configuration, thereby enhancing the efficiency of the model training process.

### 2.4. Evaluation Metrics

Four metrics were used for the evaluation of the classification performance. *Accuracy* is the ratio between the number of correctly classified samples over the total number of samples. The mathematical expression is
(5)$$Accuracy=\frac{TP+TN}{TP+TN+FP+FN},$$
where TP (*True Positive*) is the probability of a patient being correctly classified as a patient, TN (*True Negative*) is the probability of a healthy subject being correctly classified as healthy, FP (*False Positive*) is the probability of a healthy subject being incorrectly classified as a patient, and FN (*False Negative*) is the probability of a patient being incorrectly classified as healthy [26].

Additionally, we calculated the weighted average precision ($P_{WA}$), recall ($R_{WA}$), and f1 ($f1_{WA}$) score metrics, mathematically defined as
(6)$$P_{WA}=\frac{\sum_{i=1}^{n}|y_{i}|\frac{TP_{i}}{TP_{i}+FP_{i}}}{\sum_{i=1}^{n}|y_{i}|},$$
(7)$$R_{WA}=\frac{\sum_{i=1}^{n}|y_{i}|\frac{TP_{i}}{TP_{i}+FN_{i}}}{\sum_{i=1}^{n}|y_{i}|},$$
(8)$$f1_{WA}=\frac{\sum_{i=1}^{n}|y_{i}|f1score}{|y_{i}|}$$

## 3. Results

### 3.1. Experimental Results

Both methodologies described were applied to both types of signals. Through experimentation, it was determined that the Fourier transformation combined with the CNN model proved to be the optimal fit for the motor signals. This approach demonstrated effectiveness in cases where distinguishing between patient and control signals was challenging, as the transformation to the frequency spectrum enhanced the visibility of distinctive features.

Conversely, for sensory signals, which were more easily distinguishable among the three categories, the wavelet transformation in combination with the statistical models emerged as the most suitable fit. The choice of this methodology was guided by the inherent characteristics of the sensory signals, making them readily distinguishable, and the complementary strengths of the wavelet transformation and statistical model for feature extraction and classification, respectively.

Recognizing the limited size of our training dataset, consisting of a relatively small number of recordings, and aiming to mitigate the potential impact of overfitting, we adopted *k*-fold cross-validation [27]. In this methodology, the dataset was partitioned into *k* subsets. The training and testing process was iteratively conducted *k* times, with each iteration utilizing one of the k subsets as the testing set and the remaining subjects as the training set. Importantly, each subset was employed exactly once as the training set. To ascertain the overall accuracy, the average accuracy across these *k* repetitions was computed and reported. This approach ensured a robust evaluation of our model’s performance on the limited dataset.

#### 3.1.1. Motor Nerve Methodology

In the first proposed methodology, the motor train dataset was transformed into a set of images that was used to train the CONV2D model. The achieved average training accuracy, with the use of five-fold cross-validation, was 90.96%. In order to compare the capabilities of the methodology when using the proposed CONV2D architecture, we opted to try the methodology using some additional deep learning models. The models were ResNet [28], VGG-16 [29], InceptionV3 [30], and a two-dimensional artificial neural network (2D-ANN) [31]. For all additional models, we tried to keep the same architecture as the CONV2D. The results of the experiments can be seen in Table 6. We proceeded to use the best-scoring algorithm (CONV2D) to predict the classes of the 317 test samples. Additionally, we provide the confusion matrix for a visual representation of the classification procedure in Figure 7.

Based on the predictive performance of the model, it can be inferred that the classification model is reliable, as it was able to classify 298 out of 317 test samples correctly, with an accuracy score of 94%. The result suggests that the model has a high level of accuracy in its classification task, which provides confidence in its ability to generalize to future data.

#### 3.1.2. Sensory Nerve Methodology

In the second proposed methodology, we trained the five aforementioned classifiers using the set of extracted features. The results yielded the averaged accuracy score from each classifier when using the five-fold cross-validation method. In particular, the RF classifier had accuracy of 98.3%, the GB classifier scored 85%, the MLP classifier scored 78.3%, the LR classifier scored 76.6%, and the K-NN classifier had accuracy of 75.3%. From the results, we can conclude that the RF classifier had the best overall score among the models. We used the trained RF classifier in order to predict the categories of the 139 test subjects. The results of the predictions can be seen in Table 7. Additionally, we provide the confusion matrix for a visual representation of the classification procedure, using the best-scoring algorithm, in Figure 8.

Based on the classification results, it can be inferred that the proposed methodology effectively performs two key tasks. Firstly, it accurately identifies the severity of CTS. Secondly, it successfully excludes other mononeuropathies that may impact the nerves under investigation.

### 3.2. Comparison with Other Existing Approaches

We compared the proposed methodologies’ performance to that of previous CTS classification attempts. Since these works had a different number of test and training sets, it would have been unfair to directly compare them with the accuracy itself. However, we wished to evaluate the performance of our two proposed methodologies using previous works as a reference. Our proposed methodologies outperformed them by introducing two of the most efficient signal processing tools, STFT and MWD. According to Table 8, the methodology of MWD with electrodiagnostic feature extraction (EFE) using the RF classifier yielded the best results in terms of accuracy.

Park (et al., 2021) [32] used an extreme gradient boosting algorithm (XGB) in order to classify patients into three categories. They highlighted that the XGB model was superior to other machine learning models in terms of building prediction models based on regression or classification. Alcan (et al., 2020) [33] used a counter-propagation artificial neural network (CPANN) derived from the Kohonen SOMs in order to create a classifier capable of identifying CTS in patients and categorize them into four different classes. Finally, in our previous work, we extracted novel features (NFE) from sensory signals and trained a support vector machine (SVM) classifier for the CTS diagnosis problem [6], but the current approach further enhances the diagnostic accuracy.

## 4. Discussion

CTS is an entrapment neuropathy that affects a significant proportion of the population. In this study, we developed two novel methodologies using machine learning, which incorporated discrete wavelet decomposition and the short-time Fourier transform to automatically diagnose CTS in patients. The proposed motor methodology possessed some significant limitations regarding the overall diagnostic procedure, including (a) the insufficiency of motor signals to accurately evaluate the severity of a mononeuropathy and (b) the difficulty in excluding other neuropathies based exclusively on motor signals, since sensory signals are more sensitive to early changes. Therefore, to address these challenges, we introduced a second methodology, relying on the sensory signals, despite the problems arising from the low amplitude and the high noise of this signal.

Our results indicated that the developed models outperformed previous attempts at diagnosing the disease in the literature [34]. The superiority of the proposed techniques can be attributed to the efficient feature extraction process, which involved the extraction of essential features from the wavelet coefficients and frequency spectra, respectively. Furthermore, the developed models were capable of handling the highly nonlinear and complex relationships between the CTS features and the diagnosis. These results suggest that the developed models can be utilized to help health professionals to diagnose CTS more accurately and efficiently.

This study contributes significantly to the broader field of automated neuropathy diagnosis in several ways. Our two proposed methodologies have demonstrated effectiveness in accurately diagnosing signals with substantial noise, surpassing the accuracy of diagnoses made by physicians. Moreover, these methodologies offer a generalizable approach applicable to a variety of similar signals, enabling the diagnosis of various neuropathies beyond the specific focus of this study.

However, further research is necessary to investigate the models’ performance in large-scale clinical settings and to explore their effectiveness in different populations with varying degrees of severity. The proposed techniques can provide a promising foundation for the development of effective and reliable automated diagnosis systems for CTS.

## 5. Conclusions and Future Work

In conclusion, this study provides evidence for the feasibility of automatically classifying patients with CTS using two novel approaches. Our results demonstrate that the proposed methodologies outperform existing methods in terms of accuracy and precision, indicating that they could have important implications for the clinical diagnosis of CTS. The successful application of machine learning techniques in this area highlights the potential for further advancements in the use of computational methods in the field of medical diagnosis.

The results of this study contribute to the growing body of research on the use of computational methods in the diagnosis and management of nerve disorders and provide a foundation for future work aimed at improving the accuracy and reliability of automatic diagnosis in this field. In particular, these techniques hold great promise for the diagnosis of other nerve disorders, such as peripheral neuropathies, radiculopathies, and mononeuropathies.

The use of Transformers represents a promising avenue in our research agenda. While acknowledging the significance of this advanced methodology, our current work was directed towards a comprehensive exploration of more traditional approaches. This deliberate approach allowed us to establish a strong foundation by thoroughly understanding the nuances of the established methodologies, before immersing ourselves in the intricacies of Transformer models.

Further research is needed to explore the generalizability of our findings to other nerve disorders and to develop new algorithms that can accurately diagnose a wider range of conditions. By leveraging the power of artificial intelligence, we have the potential to revolutionize the way in which nerve disorders are diagnosed and managed, ultimately leading to improved patient outcomes.

## Figures and Tables

**Figure 1 bioengineering-11-00175-f001:**
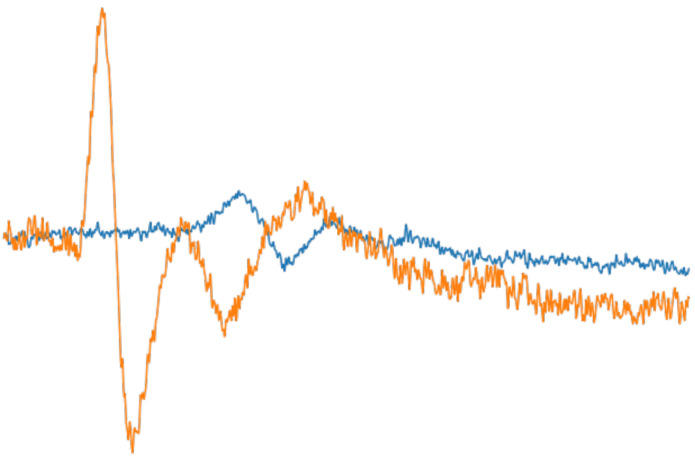
Depiction of two sensory signals from a patient with severe symptoms of CTS. The orange-colored signal was extracted from an ulnar nerve, while the blue-colored signal was extracted from a median nerve.

**Figure 2 bioengineering-11-00175-f002:**
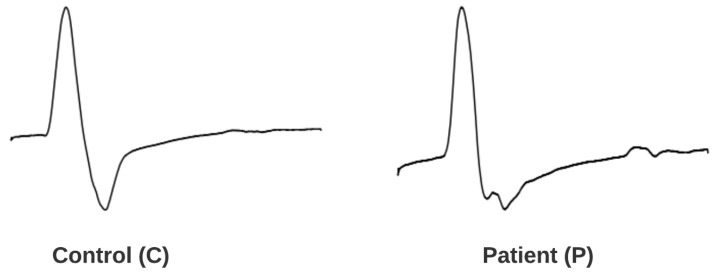
The waveform of a standard motor signal from the median nerve, contrasted with that of a motor signal affected by CTS.

**Figure 3 bioengineering-11-00175-f003:**
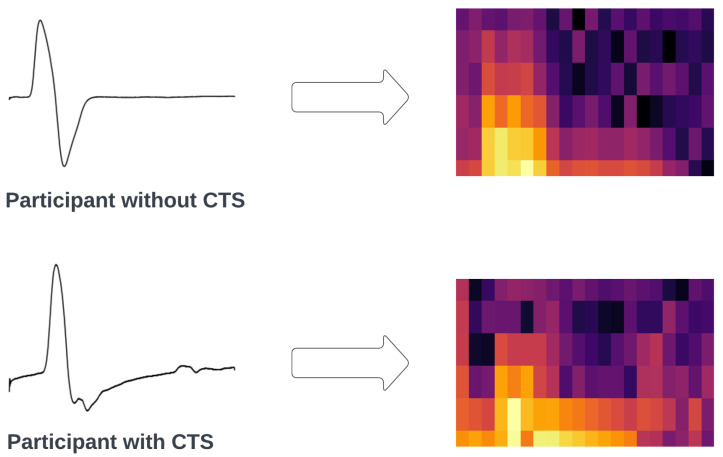
Spectrograms of two participants belonging to each class of the classification problem. The arrow indicates the transformation of the signal when applying STFT to it, while the different colors of the spectrogram indicate the amplitude of a particular frequency at a particular time.

**Figure 4 bioengineering-11-00175-f004:**
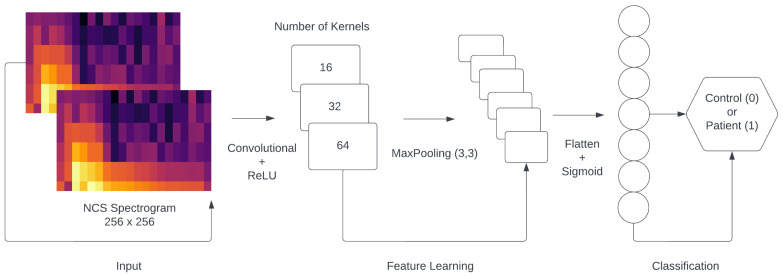
The architecture of the CONV2D classification model.

**Figure 5 bioengineering-11-00175-f005:**
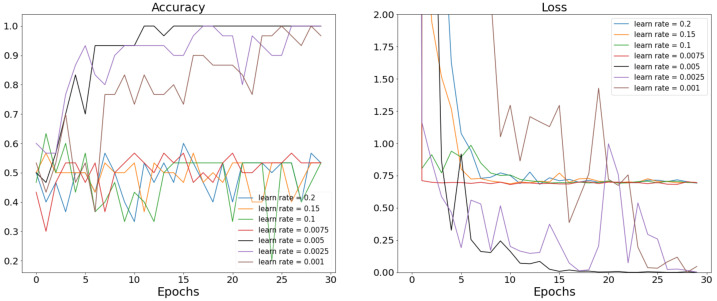
Accuracy and loss value curves of the proposed COVN2D model with different values of learning rate.

**Figure 6 bioengineering-11-00175-f006:**
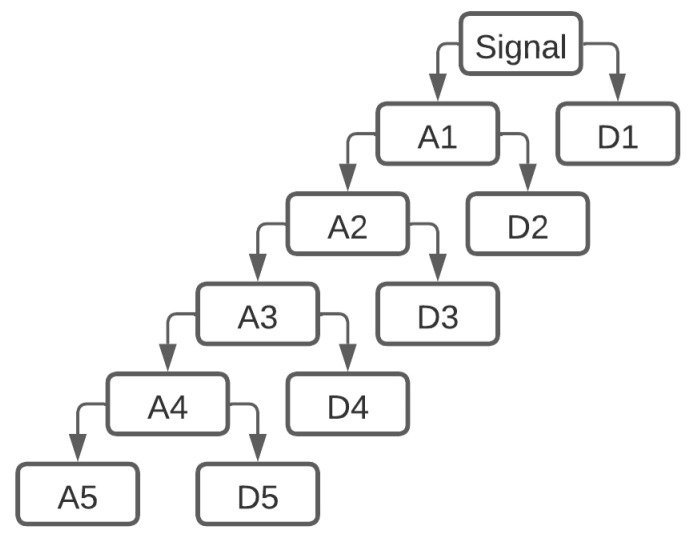
Multilevel discrete wavelet decomposition of a signal with a depth value of 5.

**Figure 7 bioengineering-11-00175-f007:**
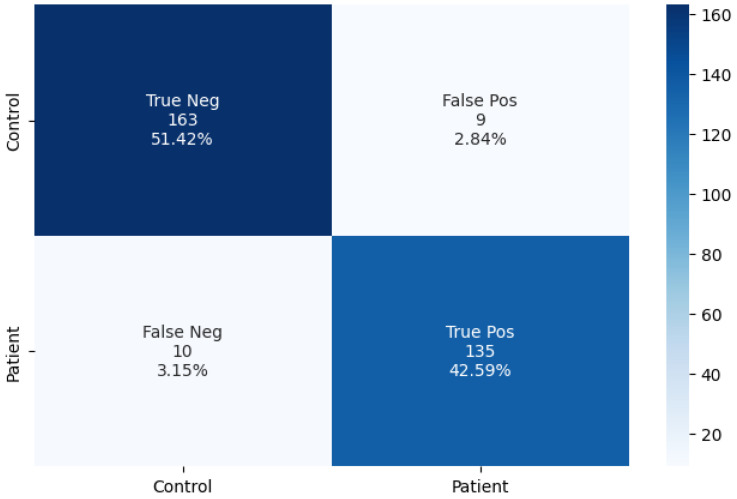
The confusion matrix of the two-class prediction.

**Figure 8 bioengineering-11-00175-f008:**
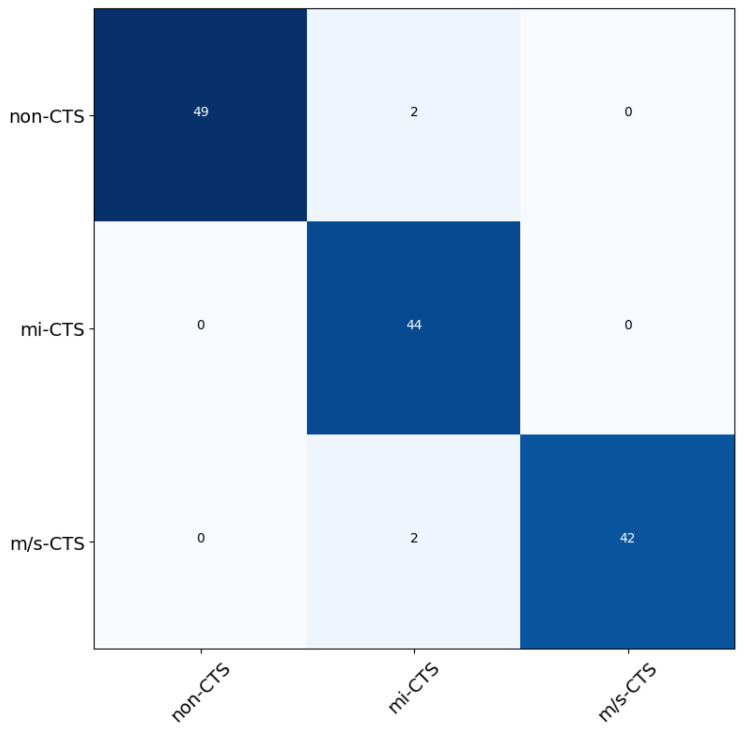
The normalized confusion matrix of the best result achieved, using the random forest (RF) classifier.

**Table 1 bioengineering-11-00175-t001:** Presentation of the first training dataset, consisting of sensory signals.

Signal	Nerve	non-CTS	mi-CST	m/s-CTS	Total
Sensory	Median/Ulnar	38	42	72	152

**Table 2 bioengineering-11-00175-t002:** Presentation of the second training dataset, consisting of motor signals.

Signal	Nerve	Control	Patient	Total
Motor	Median	38	114	152

**Table 3 bioengineering-11-00175-t003:** Presentation of the first validation dataset, consisting of sensory signals.

Nerve	non-CTS	mi-CST	m/s-CTS	Total
Median/Ulnar	51	44	44	139

**Table 4 bioengineering-11-00175-t004:** Presentation of the second validation dataset, consisting of motor signals.

Nerve	Control	Patient	Total
Median	172	145	317

**Table 5 bioengineering-11-00175-t005:** Presentation of applied methods and functions in the proposed CONV2D with explanations.

Function	Details
Convolution2D	A sliding convolution window to a two-dimensional input matrix.
ReLU	Performs linear rectification activation to an input vector of a neural network layer and outputs nonlinear results.
MaxPooling2D	Selects the highest values from a spatial domain signal given an input window.
Flatten	Transforms the multidimensional output of a convolutional layer into a one-dimensional array.
Dropout	Regularization method that partially deactivates a given percentage of the nodes from a selected hidden layer, preventing any possible scenarios of overfitting.
Dense	Most commonly used layer in machine learning. It consists of nodes that are directly connected to their preceding layer.
Sigmoid	An activation function with output values ranging from 0 to 1, suitable for binary classification problems.

**Table 6 bioengineering-11-00175-t006:** Metrics from the classification procedure, using the motor nerve conduction methodology.

Model	Accuracy	Precision	Recall	f1
CONV2D	94.00%	94.00%	94.00%	94.00%
ResNet	92.74%	92.74%	92.74%	92.74%
2D-ANN	91.79%	91.88%	91.79%	91.80%
VGG-16	90.53%	90.55%	90.53%	90.54%
InceptionV3	88.64%	88.64%	88.64%	88.64%

**Table 7 bioengineering-11-00175-t007:** Metrics from the classification procedure, using the sensory nerve conduction methodology.

Model	Accuracy	Precision	Recall	f1
RF	97.12%	97.36%	97.12%	97.15%
GB	93.52%	94.62%	93.52%	93.65%
MLP	91.36%	92.04%	91.36%	91.80%
LR	84.89%	86.09%	84.89%	85.22%
K-NN	83.45%	85.28%	83.45%	75.22%

**Table 8 bioengineering-11-00175-t008:** Comparison of classification performance with other existing methodologies.

Method	Work	Accuracy (%)
XGB	Park et al. [32]	76.6
CPANN	Alcan et al. [33]	86.6
NFE + SVM	Tsamis et al. [6]	91.1
STFT + CONV2D	Proposed	94
MWD + EFE + RF	Proposed	97.1

## Data Availability

The data presented in this study are available on request from the corresponding author.

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
