# Peer review of "Carpal Tunnel Syndrome Automated Diagnosis: A Motor vs. Sensory Nerve Conduction-Based Approach"

_bioengineering, 2024, doi:10.3390/bioengineering11020175_

Round 1

Reviewer 1 Report (Previous Reviewer 1)

Comments and Suggestions for Authors

1. There is no Introduction in the manuscript.

Introduction has been added to the manuscript

Reply: Thank you for adding the introduction. It is well written.

2. The terminology is incorrect - SNAP (sensory nerve action potential); CMAP (compound motor nerve action potential). I suggest that the manuscript is reviewed by neurophysiologist. The corresponding author of the manuscript is neurophysiologist. The terms SNAP and CMAP refer to the sensory and motor signals, respectively. To keep the terminology simple and the manuscript accessible to a diverse readership we preferred to use the terms sensory and motor signals throughout the text. However, following the reviewer’s comment, in order to align with the medical

bibliography, we have also added the terms SNAP and CMAP in the introduction (lines 41 and 50).

Reply: I still prefer using the “neurophysiologically correct” terminology of SNAP and CMAP. However, using those terms in parallel it seems a good compromise.   

3. Was the sample size calculated? I doubt that 28 patients and 10 healthy subject is sufficient number. Furthermore, the data set was split into three groups.

In the revised version of the manuscript we have significantly increased the sample size, to more than 300, according to the reviewers’ comments.

Reply: Thank you for increasing the sample size. This was the major drawback of the study. In my opinion, the Data acquisition and selection is now much more understandable.

4. Using both hands from one subject is, in this particular case, statistically incorrect. These are dependent variables rather than independent.

There is a bias in the statistical analysis when both hands of a patient are included. However, the excellent performance of the algorithm in the classification of the validation data set shows that any problem caused by this was insignificant.

Reply: Thank you for the explanation. This should be shortly explained in methods section.

5. What were the inclusion criteria for patients? Symptoms typical for CTS? Symptoms confirmed with EDx studies?

We have added in line 91 that the cases were selected randomly among recent cases with NCS in the upper extremities. This approach increases the validity of the study, since no cases were excluded.

Reply: I agree that random selection increases the validity. However, I still miss the clearly written criteria for diagnosing CTS – which criteria were chosen to establish the diagnosis of CTS and degree of compression (mild, moderate or severe). If I understand correctly, in the process of machine learning, data obtained from patients with clear criteria for CTS are presented to computer, which can than learn to recognise the typical patterns. Please supplement if I am right and ignore if I am wrong.

6. Why testing both data sets (CMAPs and SNAPSs) independently? In clinical practice, for CTS diagnosis, we perform median motor nerve conduction studies AND comparison of median and ulnar

sensory nerve conduction studies. Only in mild CTS the CMAP latency is normal. In moderate CTS is prolonged.

Further development of machine learning techniques in the analysis of signals from NCS needs independent testing of every kind of signal. So we believe that in this way our study will help others towards this direction.

Reply: Thank you for the explanation.

7. Why is proposed algorithm better than established protocols? Is faster? More reliable? More specific?

The high accuracy reached in an automatic way shows that the proposed method is better than the established protocols.

Reply: Thank you for the explanation.

Author Response

We would like to thank the reviewer for their time and effort to review our paper. Their comments have improved our manuscript and increased the scientific level. Please find below a summary of the modifications according to comments 4 and 5.

  1. Our Previous Reply: Using both hands from one subject is, in this particular case, statistically incorrect. These are dependent variables rather than independent. There is a bias in the statistical analysis when both hands of a patient are included. However, the excellent performance of the algorithm in the classification of the validation data set shows that any problem caused by this was insignificant.

Reviewer’s Reply: Thank you for the explanation. This should be shortly explained in “methods” section.

Our Reply: This aspect has been further elucidated in the manuscript to provide a more comprehensive explanation, as suggested. Please, refer to lines 84-87.

  1. I agree that random selection increases the validity. However, I still miss the clearly written criteria for diagnosing CTS – which criteria were chosen to establish the diagnosis of CTS and degree of compression (mild, moderate or severe). If I understand correctly, in the process of machine learning, data obtained from patients with clear criteria for CTS are presented to computers, which can then learn to recognize the typical patterns. Please supplement if I am right and ignore if I am wrong.

Our Reply: Thank you very much for the comment, we absolutely agree. We add in lines 96-98 the grading scale that we used for grouping all the signals.

Reviewer 2 Report (New Reviewer)

Comments and Suggestions for Authors

The authors compared sensor-based and Motor signaling for automatic identification of CTS using machine learning and deep learning methods. There is some research value, but there are more concerns that must be addressed.

1.      What is the motivation for comparing sensor and Motor signals? It is not clearly described in the manuscript.

2.      Why does the Motor solution use CNNs to classify while the Sensor solution uses traditional machine learning classifiers to classify, and is this a convincing comparison?

3.      Just for comparison, what is your technical contribution? It's more like writing up an experiment as if you've done one.

4.      Now that Transformer-based recognition models have become mainstream, authors should describe the advantages and disadvantages of comparing them with the Transformer approach in the introduction.

5.      The amount of data is too small, and k-fold cross-validation should be performed to prove the effectiveness of the method. Otherwise, 97% effectiveness was achieved in more than 100 cases, and this CTS recognition seems to be no research difficulty.

6.      The CNN methods used by the authors are also relatively old, and some SOTA CNN models should be described in the introduction to compare the strengths and weaknesses of the authors' methods with those methods. 

7.      The method of comparison is too old, and some models from the last 3-5 years should be compared.

8.      The authors use Fourier Transform to bring up features in Motor and Wavelet Transform in Sensor Methods, so comparing the effectiveness of the two methods is not convincing. Please control the variables for ablation experiments.

Comments on the Quality of English Language

None

Author Response

We would like to thank the reviewer for their time and effort to review our paper. Their comments have improved our manuscript and increased the scientific level. Please find below a summary of the modifications according to the comments.

  1. What is the motivation for comparing sensor and Motor signals? It is not clearly described in the manuscript.

Our Reply: The reason for testing and analyzing both signals is being highlighted in the Introduction section as: “Motor signals contain a comparatively lower amount of information than sensory signals, which renders the classification process quite challenging. However, these signals comprise significantly less noise than sensory signals. Thus, it is important to devise an efficient diagnostic protocol for CTS that can be carried out on motor signals.”. Since the previous sentences were not clear and we caused some confusion to the reviewer, we added additional information, in the Introduction section, about this specific choice in order to make this part more coherent to the reader. Please, refer to lines 52-59.

  1. Why does the Motor solution use CNNs to classify while the Sensor solution uses traditional machine learning classifiers to classify, and is this a convincing comparison?

Our Reply: The "Motor" methodology employs custom two-dimensional models, such as CONV2D and VGG-16, owing to the nature of the input data. The initial motor signals underwent transformation into 2D images, specifically spectrograms, rendering traditional classifiers like SVM and K-NN unsuitable for supporting this type of data.

In contrast, the "Sensory" methodology involved the extraction of features from the initial sensory signals, which were subsequently input into various traditional classifiers, including Random Forest and Multi-Layer Perceptron. While we experimented with one-dimensional deep learning models such as 1D-ANN and 1D-RNN, the obtained results did not closely align with those achieved using the aforementioned two-dimensional models. Consequently, these one-dimensional models were omitted from the manuscript, and we opted to showcase the top five performing models.

To provide the reviewer and the readers with a more comprehensive understanding of the specific choices made, additional information regarding the final decision for each methodology's application to the respective type of signal has been incorporated in lines 292-303 of the manuscript.

  1. Just for comparison, what is your technical contribution? It's more like writing up an experiment as if you've done one.

Our Reply: This study contributes significantly to the broader field of automated neuropathy diagnosis in several ways. Our two proposed methodologies have demonstrated effectiveness in accurately diagnosing signals with substantial noise, surpassing the accuracy of diagnoses made by physicians. Moreover, these methodologies offer a generalizable approach applicable to a variety of similar signals, enabling the diagnosis of various neuropathies beyond the specific focus of this study. Please, refer to lines 381-386.

  1. Now that Transformer-based recognition models have become mainstream, authors should describe the advantages and disadvantages of comparing them with the Transformer approach in the introduction.

Our Reply: We express our sincere gratitude to the reviewer for their insightful comments. Testing Transformers is indeed within our plans. However, our current focus lies in conducting a thorough investigation of more traditional approaches before delving deeply into this advanced methodology. In acknowledgment of this intention, a statement outlining our plans has been incorporated into the "Conclusion and Future Work" section of the manuscript. Please refer to lines 405-410.

  1. The amount of data is too small, and k-fold cross-validation should be performed to prove the effectiveness of the method. Otherwise, 97% effectiveness was achieved in more than 100 cases, and this CTS recognition seems to be no research difficulty.

Our Reply: Given the relatively "small" size of our dataset, we implemented k-fold cross-validation, as discussed in the Experimental Results subsection. To elucidate this approach for the reader, additional information is provided to offer a clearer understanding of the rationale behind employing k-fold cross-validation in the context of our study. Please, refer to lines 304–312.

  1. The CNN methods used by the authors are also relatively old, and some SOTA CNN models should be described in the introduction to compare the strengths and weaknesses of the authors' methods with those methods.

Our Reply: In response to this comment, we have incorporated additional references to analogous studies in order to provide a comprehensive comparison of the strengths and weaknesses of our proposed methodologies. Please, refer to lines 35-51.

  1. The method of comparison is too old, and some models from the last 3-5 years should be compared.

Our Reply: Indeed, the models compared for both methodologies were initialized a few years ago. However, their capabilities remain at the forefront, as evidenced by their continued utilization in recent studies addressing similar medical diagnostic challenges. Moreover, these models are regularly being updated with new data, ensuring their relevance and effectiveness. In our approach, transfer learning was employed to adapt the referenced pre-trained deep learning models to our specific classification objectives.

It is worth noting that the inclusion of additional models was intentionally limited to prevent the study from deviating towards an exhaustive exploration of various models. The primary focus was on the methodology itself, which led to the adoption of deep learning. This strategic decision aimed to maintain the study's coherence and emphasize the overarching methodology rather than becoming overly immersed in an extensive evaluation of individual models.

Indicative, three additional studies of the same problem are being referenced to the “Introduction” namely [3],[4] & [5] utilizing more recent CNN models. However, our proposed methodologies overcome, in accuracy, the discussed ones.

  1. The authors use Fourier Transform to bring up features in Motor and Wavelet Transform in Sensor Methods, so comparing the effectiveness of the two methods is not convincing. Please control the variables for ablation experiments.

Our Reply: Both methodologies were employed for both types of signals. Nevertheless, the intrinsic characteristics of these signals, coupled with a series of tests conducted, reveal that the Fourier approach exhibited superior performance when applied to motor signals, where differentiation between control and patient groups was challenging. Conversely, the Wavelet approach proved to be more effective for sensory signals, particularly in the presence of substantial noise, facilitating the extraction of electrodiagnostic features. In summary, our findings demonstrate that, for each type of signal, the most fitting approach is showcased based on its ability to address the unique challenges associated with motor and sensory data.

As articulated in our response to the 2nd comment from the reviewer, which is closely related to the current one, in order to provide a more comprehensive understanding of the specific choices made, additional information regarding the final decision for each methodology's application to the respective type of signal has been incorporated in lines 292-303 of the manuscript.

Round 2

Reviewer 2 Report (New Reviewer)

Comments and Suggestions for Authors

The authors have done great work to address my concerns.

Comments on the Quality of English Language

None

This manuscript is a resubmission of an earlier submission. The following is a list of the peer review reports and author responses from that submission.

Round 1

Reviewer 1 Report

Comments and Suggestions for Authors

The objective of study on Carpal Tunnel Syndrome Automated Diagnosis: A Motor vs Sensory Nerve Conduction based Approach was to evaluate the effectiveness of machine learning classification techniques applied to nerve conduction studies motor and sensory signals for the automatic diagnosis of carpal tunnel syndrome (CTS). The results show that both methodologies surpassed previous attempts for the automatic CTS diagnosis. The manuscript is not well written and methodology is flawed. It also has some other problems described in comments bellow.

1. There is no Introduction in the manuscript.

2. The terminology is incorrect - SNAP (sensory nerve action potential); CMAP (compound motor nerve action potential). I suggest that the manuscript is reviewed by neurophysiologist.

3. Was the sample size calculated? I doubt that 28 patients and 10 healthy subject is sufficient number. Furthermore, the data set was split into three groups.

4. Using both hands from one subject is, in this particular case, statistically incorrect. These are dependent variables rather than independent.

5. What was the inclusion criteria for patients? Symptoms typical for CTS? Symptoms confirmed with EDx studies?

6. Why testing both data sets (CMAPs and SNAPSs) independently? In clinical practice, for CTS diagnosis, we perform median motor nerve conduction studies AND comparison of median and ulnar sensory nerve conduction studies. Only in mild CTS the CMAP latency is normal. In moderate CTS is prolonged.

7. Why is proposed algorithm better than established protocols?1 Is faster? more reliable? More specific?

Receive my kind regards

1. Zivkovic S, Gruener G, Arnold M, Winter C, Nuckols T, Narayanaswami P; the Quality Improvement Committee of the American Association of Neuromuscular & Electrodiagnostic Medicine. Quality measures in electrodiagnosis: Carpal tunnel syndrome-An AANEM Quality Measure Set. Muscle Nerve. 2020 Apr;61(4):460-465. doi: 10.1002/mus.26810. Epub 2020 Feb 3. PMID: 31950523.

Comments on the Quality of English Language

The quality of English language must be improved. The sentences are hard to read and follow. 

Reviewer 2 Report

Comments and Suggestions for Authors

This study addresses an important clinical problem of diagnosing the presence and severity of CTS and shows that AI (CNN and MWD) can markedly increase the diagnostic accuracy for detecting CTS.

1.        Did the patients have any specific etiology for their CTS and if so, can the AI methods differentiate these etiologies i.e. if this was due to diabetes etc.?

2.        I am surprised that the authors had access to such as small number of tests and validation data sets. Neurophysiological laboratories all over the world literally have thousands of patient data sets.

3.        What proportion of the one dimensional signals could be transformed?

Reviewer 3 Report

Comments and Suggestions for Authors

The authors have proposed the convolutional neural network (CNN) to evaluate the effectiveness of machine learning classification techniques through the application of nerve conduction studies motor and sensory signals for the automatic diagnosis of carpal tunnel syndrome in the medical field. This study is interesting however there are some drawbacks that the authors should address them to improve this study.

  1. The introduction is lacked in this study as well as the literature review part.
  2. The scientific basis of selected methodologies must be indicated in this study. Why the machine learning/CNN has been employed to predict this disease.
  3. In the discussion, the authors must analyzed clearly the strength points of this proposed method with other machine learning techniques like ANN, YoLo V5, 7 etc.
  4. The acknowledgement should be implemented in this study.
Comments on the Quality of English Language

The language in English should be improved in this paper.